# Work in Progress: Enhancing Human-Robot Interaction through a Speech and Command Recognition System for a Service Robot Using ROS Melodic

*Note: Sub-titles are not captured in Xplore and should not be used

Luis Emiliano Rodríguez Raygoza
*Tecnologico de Monterrey*
*School of Engineering and Sciences*
Monterrey, México
a01252086@tec.mx

Jorge De-J. Lozoya-Santos
*Tecnologico de Monterrey*
*School of Engineering and Sciences*
Monterrey, México
jorge.lozoya@tec.mx

Luis C. Félix-Herrán
*Tecnologico de Monterrey*
*School of Engineering and Sciences*
Monterrey, México
lcfelix@tec.mx

Juan C. Tudon-Martinez
*Tecnologico de Monterrey*
*School of Engineering and Sciences*
Monterrey, México
jc.tudon@tec.mx

*Abstract*—This paper presents the development and evaluation of a Speech and Command Recognition system integrated into PiBot, an autonomous service robot developed at Tecnológico de Monterrey. The system executes on Robot Operating System (ROS) Melodic framework running on a Jetson TX2 embedded computer to enable natural language interaction through Automated Speech Recognition (ASR). The study focuses on the challenges and opportunities of implementing speech recognition in real-world environments, particularly within constrained hardware platforms. The system achieved a 25% Word Error Rate (WER) and a 73% Command Accuracy, with performance varying across different testing environments. The system achieved a 25% Word Error Rate (WER) and a 73% Command Accuracy, with performance varying across different testing environments. Difficulties were noted in recognizing uncommon or non-Spanish words. A comparison with state-of-the-art models indicates room for improvement. Future work will focus on fine-tuning the model using datasets with ground truth transcriptions to enhance reliability in complex, noise-prone settings.

*Index Terms*—Automated Speech Recognition (ASR), Human-Robot Interaction (HRI), Service Robots, Command Detection, Embedded Systems

## I. INTRODUCTION

In recent years, robotics has made notable progress, with service robots becoming prominent solutions designed to communicate, interact, and assist customers [8]. As society moves toward greater automation, effective human-robot interaction is increasingly important. Among the key elements facilitating this interaction, speech algorithms are essential tools and widely used approaches in Human-Robot Interaction (HRI) [7]. Speech functions both as an input and an output in dialogue systems. As an input, it allows robots to recognize spoken language through Speech-to-Text (STT) or Automated Speech Recognition (ASR). As an output, speech synthesis converts textual responses into spoken language, enabling natural language interaction [1].

This paper presents preliminary work focusing on the development and evaluation of these systems to identify areas for future improvement. The system is integrated into PiBot, an autonomous service robot developed at Tecnológico de Monterrey, with its design and development previously described [5]. PiBot's algorithms run within the Robot Operating System (ROS) Melodic framework on Ubuntu 18.04, utilizing the processing capabilities of a Jetson TX2 embedded computer. While this combination of software and hardware is functional, it presents limitations due to the constraints of embedded computer architecture, reliance on battery power, and limited availability of GPU-accelerated library versions. Additionally, the Jetson TX2, an older model, poses specific challenges impacting the system's performance and flexibility.

This paper examines the challenges and opportunities of integrating speech recognition algorithms within a service robot, emphasizing the practical implementation and evaluation of these systems in real-world settings. Through experimentation and analysis, we aim to identify the strengths and limitations of current speech recognition technologies when deployed on constrained hardware platforms like the Jetson TX2, providing insights that may inform future enhancements in human-robot interaction in complex, noise-prone environments.

The paper is organized as follows: Section II describes the

system integration and technological configuration. Section III outlines the configuration and operation of the developed processing nodes, detailing their roles in the processing pipeline. Section IV discusses the testing and validation methodology used to evaluate the system. The validation results are presented in Section V, and conclusions are drawn in Section VI.

## II. PiBot's Speech Recognition Integration

The development of the Speech and Command Recognition System for PiBot is a significant enhancement, significantly improving its interactive capabilities. This system, detailed in this section, facilitates verbal communication between humans and PiBot, extending interactions beyond the existing terminal and web interface. It also sets the stage for future voice-activated motion tasks with their respective algorithms. The integration process, which began with the ReSpeaker Mic Array, is a crucial step in this journey. This device, connected via USB, provides raw audio data and the relative direction of sound, which will be used to activate motion tasks through speech, thereby enhancing PiBot's functionality.

The implementation of the Speech and Command Recognition System for PiBot is designed to enable it to respond to vocal instructions, a common interface method for HRI. The process begins by connecting the ReSpeaker Mic Array and setting up an inference node. This node is dedicated to pre-processing the audio signal, performing inference on the audio data, and post-processing the results to obtain interpreted speech. The system then uses score criteria to determine whether a command is present in the inferred text. If a command is detected, it is forwarded to a state machine to execute the appropriate task on PiBot.

### A. Technological Framework and Adaptations

This section describes the hardware and software architecture that integrates the Speech and Command Recognition system into PiBot. The integration is supported by a Jetson TX2 embedded computer running Ubuntu 18.04 with the JetPack 4.6.5 SDK and a ReSpeaker 2.0 Mic Array connected via USB for capturing audio input. The Jetson TX2 serves as the central processing unit, handling all computational tasks including audio inference, navigation, and sensor fusion. The JetPack SDK includes essential libraries, such as CUDA 10.2 and cuDNN 8.2.1, providing GPU acceleration to handle the demanding deep learning inference tasks required for real-time operation [4].

The ROS Melodic framework provides a robust environment for developing modular nodes that handle specific tasks within the speech recognition pipeline. The Jetson TX2's GPU accelerates the inference process of the ASR model, enabling real-time speech processing. ROS topics and services are used for inter-node communication, allowing the ReSpeaker Node to publish audio data, the Inference Node to perform GPU-accelerated speech-to-text conversion, and the Command Detection Node to interpret commands. The State Machine Node orchestrates the execution of commands, leveraging

ROS's actionlib for asynchronous task handling, which ensures that multiple actions can be managed concurrently.

Due to the limitations of the Jetson TX2 hardware, several strategies were explored for configuring the necessary software environment. The Nvidia JetPack SDK is crucial for hardware-accelerated AI development, but due to compatibility constraints, the available machine learning frameworks are limited to older versions. Initially, a conda environment was considered for managing the library versions required for inference, but the lack of ARM-compatible versions proved to be a significant barrier. A compatible PyTorch Docker container was also investigated, offering GPU-accelerated support for speech recognition. Despite the potential, this approach faced practical challenges related to processing demands and frequent image deletions.

Ultimately, we installed a specific version of PyTorch (provided by Nvidia) that works with CUDA 10.2, enabling us to perform GPU-accelerated inference for speech recognition tasks. This required transitioning from the torchaudio library to the librosa library for certain audio processing tasks, maintaining the same inference approach with some modifications. The difference in performance was significant: GPU-accelerated inference took 3–4 seconds, while CPU-based inference took approximately 55 seconds, emphasizing the necessity of GPU acceleration for achieving near real-time response. Figure X illustrates the hardware and software integration within PiBot, including the flow between ROS nodes, the Jetson TX2, and the ReSpeaker Mic Array.

## III. Operation of Processing Nodes

This section details the setup and functionality of the processing nodes developed for the speech and command recognition system, highlighting their roles within the framework. The system handles audio input, speech recognition, command detection, and command execution through four distinct nodes. Each node operates within the processing pipeline, collectively ensuring the system's functionality. The initial node was adapted from an existing ROS Melodic package [2], which facilitates communication with the ReSpeaker 2.0 Mic Array. This array captures audio input and provides directional sound data using its quad-microphone setup. The directional information is intended for future enhancements, such as activating motion tasks based on speech direction.

The second node, developed specifically for this implementation, handles inference. It receives audio segments, pre-processes the data to reduce background noise, performs GPU-accelerated inference using the *jonatasgrosman/wav2vec2-large-xlsr-53-spanish* model [3], and post-processes the results to identify keywords. The third node processes the inferred text to determine if it contains a command from a predefined set of keywords and thresholds. Finally, the fourth node functions as a state machine, waiting for commands and executing the corresponding tasks on PiBot.

Figure 1 illustrates the interconnection of these nodes, the topics they broadcast, and the data types transmitted between them. This visual aid clarifies the communication flow and the

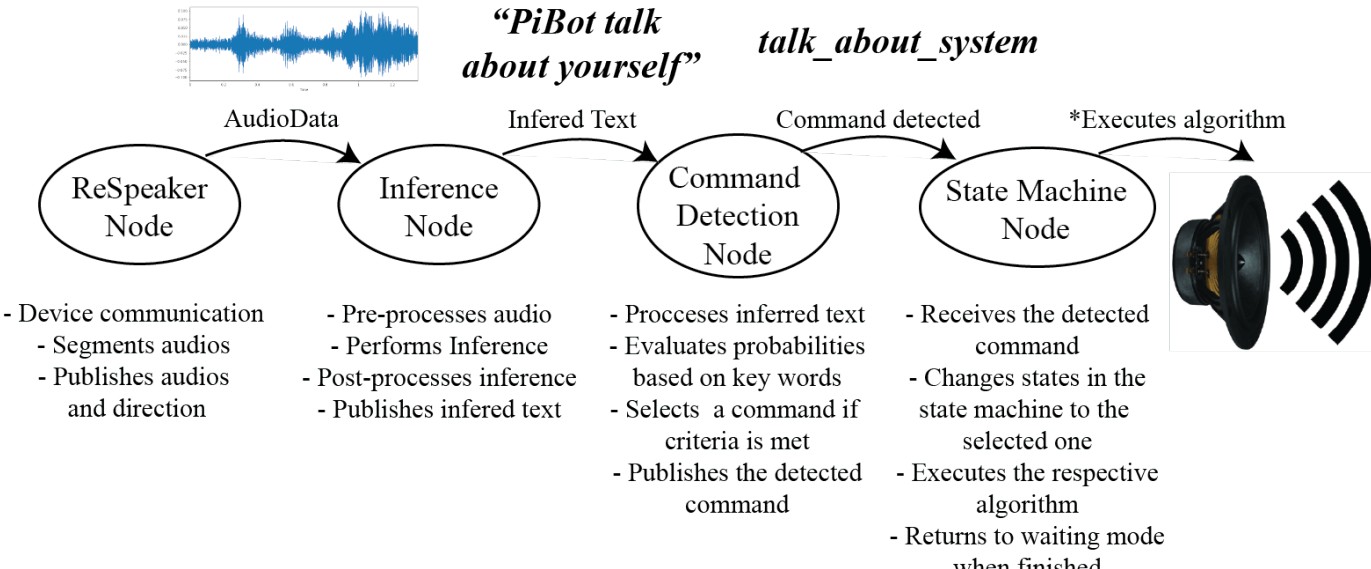

Fig. 1. Schematic representation of the audio processing framework, showcasing the workflow from audio capture to command execution. It begins with the ReSpeaker Node processing audio data, followed by the Inference Node for text inference and processing, and the Command Detection Node for command detection and selection. The State Machine Node completes the sequence by executing the corresponding actions.

sequential processing steps from one node to the next. Each node and the algorithms employed are further explained in the subsequent subsections.

### A. Inference Node Configuration and Operation

The Inference Node initiates the speech recognition process by handling audio segments, pre-processing them, performing inference, and converting the results into text. The node subscribes to the "/speech_audio" ROS topic, where audio segments are continuously published by the ReSpeaker node. Upon receiving an audio message, the data undergoes several processing steps before the inferred text is published to the "/audio_text_topic" as a String message for the next node.

Audio processing begins with the initialization of necessary libraries. The 'rospy' library facilitates communication between ROS nodes, while 'librosa' and 'soundfile' are used for audio processing. Additional libraries support array manipulation, audio transformations, and machine learning tasks. The core of the inference task utilizes the wav2vec2-large-xlsr-53-spanish model, a speech recognition model already fine-tuned with the Common Voice Corpus 6.1 dataset for Spanish. This model, sources from Hugging Face [3], operates on *wav* files sampled at 16,000 Hz. The dataset used for fine-tuning, Common Voice Corpus 6.1, provides a diverse range of transcriptions. For our implementation, we leveraged this pre-existing fine-tuned model to process our collected audio data without further modification. Global variables are set during initialization, and the GPU device is configured. A similarity threshold is established for fuzzy word matching, and a spell checker and a set of keywords are initialized to address common inference errors.

Once the model is ready, which takes approximately 20 seconds, the ROS node and subscriber are activated to lis-

ten for audio data on the "/speech_audio" topic. The main processing occurs in the callback function, which is triggered upon receiving an audio message. The audio data is converted to a .wav file and loaded into a GPU-compatible tensor. Pre-processing is performed using the 'noisereduce' library to minimize background noise, as illustrated in Figure 2, which shows the effects of noise reduction on different audio signals.

The filtered audio is then processed by the Wav2Vec 2.0 model, which transcribes the spoken content into text. Post-processing involves mapping the inferred text to correct common transcription errors. This includes mapping terms like 'piot' to 'pibot', 'pivot' to 'pibot', and 'machin' to 'machine'. Additionally, the 'fuzzywuzzy' library performs word corrections based on the Levenshtein Distance, allowing for corrections with a similarity threshold of 70%. Keywords such as 'pibot', 'pibotino', and 'patrullar' are specifically targeted for this correction process. The refined text is then published to the "/audio_text_topic" ROS topic for the Command Detection Node to process.

### B. Command Detection Node

The Command Detection Node converts inferred text into detected commands to execute later. It subscribes to the "/audio_text_topic" ROS topic to receive text inputs from the Inference Node. Upon receiving a message, the node processes the text by tokenizing it into individual words for detailed analysis. Special attention is given to the keywords "pibot" and "pibotino," which identify the robot intended to receive the commands. Detecting these keywords ensures that only relevant commands are processed, filtering out unrelated speech.

After recognizing the robot identifier, the node maps keywords associated with each potential command. These key-

| Command | Intent | Keywords | Thresholds |
|---|---|---|---|
| talk_about_system | For PiBot to play a series of audio files, explaining itself, its capabilities, its components and its features. | (hablame, háblame, cuéntame, cuentame, explicame, explícate), (ti), (sobre), (capacidades), (componentes) | 0.2 |
| talk_about_machine_care | For PiBot to play a series of audio files, explaining Machine Care, a strategical business partner for PiBot development. | (hablame, háblame, cuentame, cuéntame, explicame), (machin, machine, care, quer) | 0.6 |
| talk_about_event | For PiBot to play a series of audio files, explaining the ENCLELAC event which invited PiBot to attend. | (háblame, hablame, cuéntame, cuentame, explicame), (evento, clelac, conferencia, enclelac, claustro) | 0.6 |
| come_towards_me | Command for a future action intended to command PiBot to navigate towards the person closest to the sound direction. | (ven, vente, acercate, aproxima, aproximate), (aca, acá, aquí) | 0.6 |
| patrol | Command for future to start patrolling actions on PiBot, navigating autonomously in a set of predefined points. | (patrullaje, patrullar, vuelta), (empieza, comienza, ponte) | 0.2 |
| look_at_me | Command for future action intended to command PiBot to rotate to face the sound direction source. | (voltea, volteame, observame, observa, boltea, mirame, mira), (empieza, comienza, ponte) | 0.3 |
| stop_action | Some states are indefinite and are only stopped by this action. Additionally, the audio sequences can be stopped with this command. | (detente, alto, parate, cancela, basta, termina, interrumpe, suspende, aborta) | 0.2 |
| continue | Signals PiBot to continue with its last state, either continuing from the last played audio, or continuing an indefinite task. | (continúa, continua, reanuda) | 1 |

TABLE I
LIST OF COMMANDS PROGRAMMED IN THE SPEECH RECOGNITION IMPLEMENTATION FOR PiBot. THE LIST PRESENTS THE IDENTIFIER, THE INTENT FOR THIS STATE (IN THE STATE MACHINE), KEYWORDS FOR EACH COMMAND, AND THE MINIMUM SCORE THRESHOLD TO SURPASS TO BE CONSIDERED A POSSIBLE CANDIDATE.

words are grouped by synonyms to improve the accuracy of the scoring mechanism, which determines the most likely intended command. This grouping prevents score inflation from repetitive similar words, ensuring a more accurate interpretation. The algorithm then evaluates the scores for each potential command against predefined thresholds. If a command's score exceeds its threshold and is the highest among the candidates, it is selected for execution. The selected command is published to the "/state_topic" ROS topic as a String message for the State Machine Node to execute relevant algorithms.

Table I lists the programmed commands, their intended actions, associated keywords, and the minimum score thresholds required for selection.

### C. State Machine Node

The State Machine Node manages the execution of tasks corresponding to received commands using the *smach* library. Each state within the state machine performs specific functions and makes decisions based on the commands received. The final state, *FinishProgram*, handles the concluding logic before terminating the program. Figure 3 depicts the structure of the state machine, which begins in the *Initial Setup* state and transitions to the *Waiting Mode* state once all necessary nodes are active. Upon receiving a command, the state machine transitions to the appropriate state to execute the corresponding task and then returns to *Waiting Mode* after completion or interruption. If the state machine is stopped, it moves to the *End* state.

The *Initial Setup* state verifies that all required ROS nodes, specifically the Inference Node and Command Detection Node, are operational. This verification accounts for the approximately 20-second power-on time. Once confirmed, the state machine transitions to the *Waiting Mode* state, where it remains ready to receive and process commands from the "/state_topic" ROS topic.

In the *Waiting Mode* state, the state machine continuously monitors for incoming commands. Upon receiving a command, it transitions to the corresponding state to execute the associated actions. All states responsible for providing explanations are fully operational, playing a series of audio files as intended. Users can interrupt these actions with the 'stop_action' command, which halts audio playback and records the last played audio. The 'continue' command also allows users to resume the last interrupted action. While commands such as 'patrol', 'come_towards_me', and 'look_at_me' are recognized and processed, the actions they trigger are scheduled for future development.

Each state within the state machine is designed to handle specific functionalities, with clearly defined transitions ensuring a reliable and adaptable system. Audio feedback is integral to the state machine, enhancing user interaction by confirming received commands. When a command is successfully detected, the system plays a randomly selected confirmation audio from a pool of pre-recorded phrases. This approach confirms command recognition and adds variety to interactions, aiming to improve the user experience. Similarly, continuation commands trigger randomized audio feedback to inform users that the system has resumed its previous action.

### IV. TESTING AND VALIDATION OF SPEECH RECOGNITION IMPLEMENTATION

Evaluating the performance of the speech recognition system implemented in PiBot is essential to ensure effective interaction and accurate command execution and to highlight critical areas of opportunity for future work in this platform. This section details the testing methodology, including the audio samples, testing environments, and the metrics used

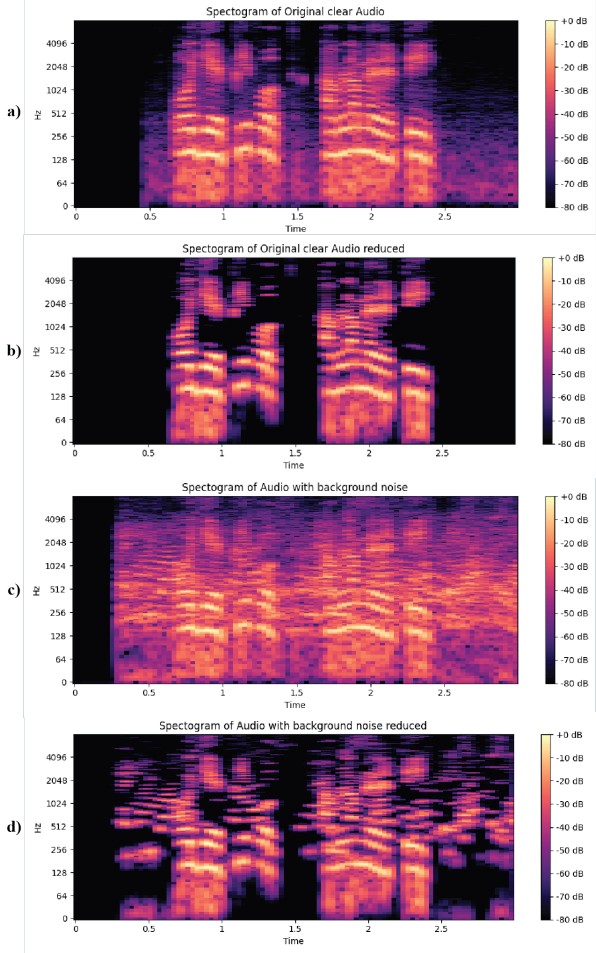

Fig. 2. Comparison of Audio Waveforms Across Various Scenarios (spoken text said is "oye pibot hablame de ti"): a) Original clear audio waveform, b) Noise-reduced waveform from the original clear audio, c) Original audio waveform with added background noise, d) Noise-reduced waveform of the audio with added background noise.

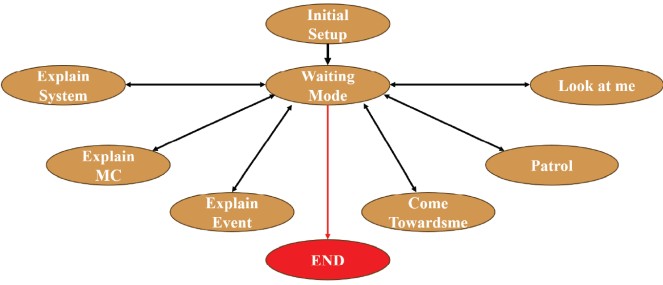

Fig. 3. Illustration of the State Machine structure implemented on PiBot, to perform different algorithms based on the command received.

| Desired Command | Transcription |
|---|---|
| look_at_me | oye pibot voltea |
| talk_about_system | oye pibot hablame de ti |
| talk_about_event | oye pibot hablame del evento |
| talk_about_event | oye pibot hablame de clelac |
| look_at_me | oye pibot voltea |
| patrol | oye pibot ponte a patrullar |
| talk_about_machine_care | oye pibot hablame de machine care |
| come_towards_me | oye pibot ven para aca |
| talk_about_system | oye pibot cuentame sobre ti |
| talk_about_event | oye pibot que sabes del evento |
| patrol | oye pibot comienza a patrullar |

TABLE II
LIST OF PHRASES WITH THE INTENDED COMMAND USED ON THE SPEECH RECOGNITION TESTING.

to assess system reliability and accuracy. The testing was performed on a computer running ROS Melodic on Windows 11 through the Windows Subsystem for Linux with Ubuntu 18.04. The focus was on evaluating the algorithmic accuracy and reliability consistent across different platforms.

### A. Recording and Preparation of Audio Data

Audio recordings for this validation were captured using PiBot's ReSpeaker Mic Array. A Python script defined each audio segment's start and end times based on keyboard inputs. This resulted in individual *.wav* files named according to the spoken phrase and the recording location; all recordings were mono-channel with a sample rate of 16,000 Hz. Some recordings included English words such as "machine" and "care" to test the system's handling of multilingual inputs even when fine-tuned with a Spanish dataset. Table II lists the phrases and their corresponding intended commands. It is important to note that all commands are in Spanish, aligning with the system's target language.

### B. Testing Environments

The speech recognition system was tested in three different environments to evaluate its performance under varying noise conditions:

- **Office Floor**: Recordings were made on the second floor of the CETEC tower at Tecnológico de Monterrey's Innovaction floor, where undergraduate students presented their final projects. This environment featured significant background noise due to multiple conversations and activities, providing a challenging setting for speech recognition.
- **Library**: PiBot was positioned on the first floor of Tecnológico de Monterrey's library. Electrical escalators, nearby shops, and student activity contributed to background noise, testing the system's ability to function in a moderately noisy environment.
- **Laboratory**: Recordings in the laboratory were conducted in a wide, open space with minimal background noise. This environment served as a control to assess the system's performance in ideal conditions.

During testing, the speaker maintained a consistent speaking volume and pace. The impact of factors such as background

noise, echo, and microphone distance were analyzed to understand their effects on WER and Command Accuracy.

Figures 4, 5, and 6 show PiBot's placement in each of these environments.

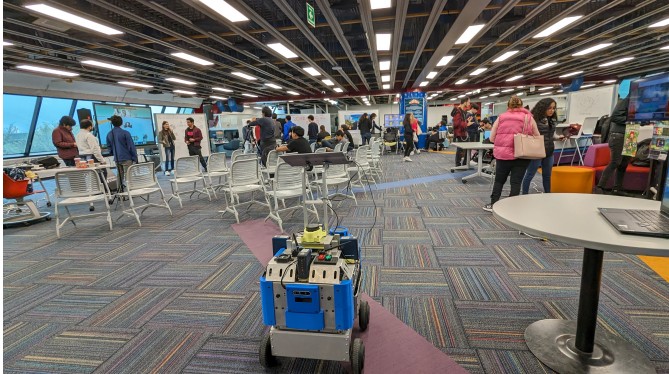

Fig. 4. PiBot located at Innovaction, where the set of phrases were recorded.

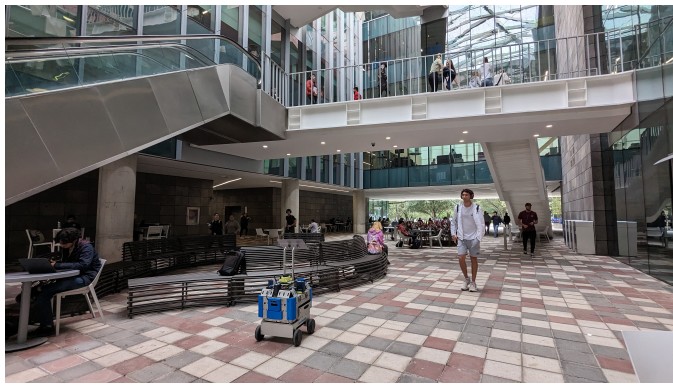

Fig. 5. PiBot located at Tec's Library, where the set of phrases were recorded.

### C. Evaluation Metrics

Two primary metrics were used to evaluate the system's performance: Word Error Rate (WER) and Command Prediction Accuracy.

*1) Word Error Rate (WER):* WER measures the difference between the recognized word sequence and the ground truth transcription by calculating the minimum number of substitutions, insertions, and deletions required to transform one sequence into the other. It is calculated using the following formula:

$$\text{Word Error Rate} = \frac{S + I + D}{N} \quad (1)$$

- **S** (Substitutions): The number of words in the recognized transcription that differ from the ground truth.
- **I** (Insertions): The number of additional words present in the recognized transcription that are not in the ground truth.
- **D** (Deletions): The number of words from the ground truth that are missing in the recognized transcription.

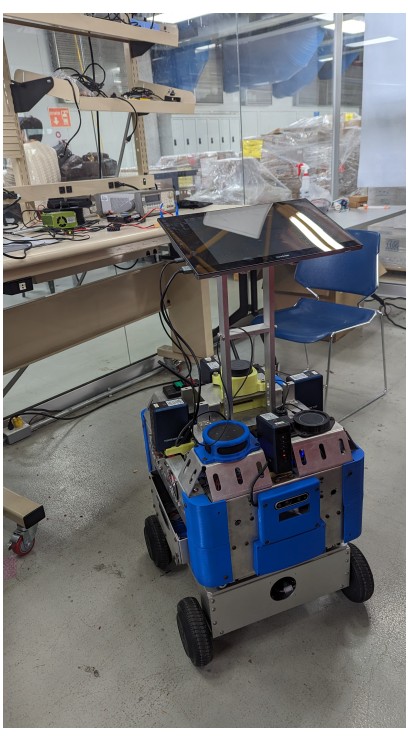

Fig. 6. PiBot located at Tec's Laboratory, where the set of phrases were recorded.

- **N** (Number of words): The total number of words in the ground truth transcription.

WER was calculated using the `jewel` Python library, which compares the inferred transcription against the ground truth.

*2) Command Prediction Accuracy:* Command Prediction Accuracy evaluates whether the system correctly identifies and executes the intended command. This metric is binary: a score of 1 is assigned if the inferred command matches the ground truth command, and a score of 0 otherwise.

## V. RESULTS

The tests were designed to evaluate the performance of the speech recognition pipeline within operational scenarios, specifically examining how spoken words trigger commands in PiBot's state machine. These evaluations assess the system's reliability and identify potential areas for enhancement. The results are detailed in Table V, which includes each file's name, inferred text, selected command, Word Error Rate (WER), and command accuracy for each scenario. Additionally, Table III summarizes the metrics to provide an overview of overall averages and location-specific performance.

|  | Word Error Rate | Command Accuracy |
|---|---|---|
| Library | 42% | 50% |
| Laboratory | 19% | 70% |
| Office | 13% | 100% |
| Average | 25% | 73% |

TABLE III
AVERAGES OF WER AND COMMAND ACCURACY AVERAGES IN DIFFERENT LOCATIONS AND OVERALL.

PiBot's speech recognition system achieved an overall command accuracy of 73% and a WER of 25%. Compared to advanced models such as OpenAI's Whisper, which achieves a WER below 9% and maintains performance in noisy environments [6], there is potential for further improvement in PiBot's system. The variation in WER and Command Accuracy across different environments suggests that factors beyond general noise levels influence system performance. In the library, the open space and echoes likely contributed to a higher WER of 42% and lower Command Accuracy of 50%. Despite high background noise in the office, the confined space may have allowed the microphone array to better capture the speaker's voice, resulting in a lower WER of 13% and high Command Accuracy. The laboratory, covered by glass walls and an open ceiling, showed intermediate results. These findings could indicate that environmental acoustics, such as echo and reverberation, and the directional characteristics of background noise, significantly impact the system's effectiveness.

## VI. Conclusions

This study evaluated a speech recognition and command detection system for the PiBot platform, achieving an average Word Error Rate (WER) of 25% and a Command Accuracy of 73%. The system's performance varied across different testing environments, with the library setting exhibiting the highest WER of 42% and the lowest Command Accuracy of 50%. Conversely, despite its high background noise, the office environment demonstrated a WER of 13% and a Command Accuracy of 100%. The laboratory environment, characterized by minimal background noise, showed a WER of 19% and a Command Accuracy of 70%.

These results might indicate that factors beyond the general noise level influence the system's performance. The unexpectedly high accuracy in the noisy office environment suggests that the system can perform well in high background noise scenarios, while the specific conditions remain unclear. In contrast, the library's moderate noise levels adversely affected both WER and Command Accuracy, being the only open location. Additionally, the system faced challenges in recognizing certain words, particularly those uncommon or not in Spanish, such as "Enclelac" and "Machine Care." This difficulty aligns with existing research, which indicates that proper nouns and less frequent terms are more susceptible to recognition errors in speech systems [1]. To address the difficulties in recognizing uncommon or non-Spanish words, we implemented post-processing techniques in the Command Detection Node, specifically mapping commonly unrecognized words to the correct terms. PiBot, the system's name, was common but had special difficulty due to the wide range of inferences.

However, due to limitations and lack of a dataset for this initial setup, these methods had limited effectiveness, particularly in complex acoustic environments. To improve this, future work should focus on fine-tuning the model using a more comprehensive dataset, which should include a wide range set of words and phrases that the system is expected to handle. This dataset should be captured using the ReSpeaker microphone array across various environments with different noise levels. Such customization will likely enhance the model's ability to recognize specific terms and increase overall command accuracy, as the current lack of effective keyword detection negatively impacts the performance of short command phrases.

Furthermore, advancing pre-processing techniques, mainly through more effective noise reduction methods, could significantly increase the system's robustness and accuracy. Some noise reduction techniques include other libraries for spectral subtraction and deep learning-based noise impression algorithms that do not affect speech recognition tasks. These improvements are essential for ensuring reliable and practical real-world applications of PiBot in various and potentially challenging acoustic environments.

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

| Filename | Inferred Text | Detected Command | WER | Command Accuracy |
|---|---|---|---|---|
| comienza_patrullar.wav | oyemiotoveja patrullar | | 0.8 | 0 |
| comienza_patrullar_laboratorio.wav | oye ven pibot comienza a patrullar | patrol | 0.2 | 1 |
| comienza_patrullar_office.wav | oye pibot comienza a patrullar | patrol | 0 | 1 |
| cuentame_laboratorio.wav | oye pibot cuentame sobre ti | talk_about_system | 0 | 1 |
| cuentame_sobre_ti_biblio.wav | oye pibot cuentame sobre ti | talk_about_system | 0 | 1 |
| cuentame_ti_office.wav | oye pibot cuentame sobre ti | talk_about_system | 0 | 1 |
| hablameenclelac.wav | oye pibot hablame de | | 0.2 | 0 |
| hablameenclelac_biblio.wav | eoundedte | | 1 | 0 |
| hablameenclelac_office.wav | oye pibot hablame de enclelac | talk_about_event | 0 | 1 |
| hablameevento_biblio.wav | oye pibot o aca del evento | talk_about_event | 0.4 | 1 |
| hablameevento_office.wav | oye pibot hablame de evento | talk_about_event | 0.2 | 1 |
| hablamedeeenclelac_laboratorio.wav | oye pibot hablame enclelac | talk_about_event | 0.2 | 1 |
| hablamedelevento_laboratorio.wav | oye pibot hablame de evento | | 0.2 | 0 |
| hablamedeti_biblio.wav | oye pibot hablame de ti | talk_about_system | 0 | 1 |
| hablamedeti_laboratorio.wav | oye pibot hablame de ti | talk_about_system | 0 | 1 |
| hablamedeti_office.wav | oye pibot hablame de ti | talk_about_system | 0 | 1 |
| mc_biblio.wav | oye pibot hablame de aca | | 0.33 | 0 |
| mc_laboratiorio.wav | oye pibot hablame de | | 0.33 | 0 |
| mc_office.wav | oye pibot hablame de machin | talk_about_machine_care | 0.33 | 1 |
| patrullar_laboratorio.wav | oye mibun patrullar | | 0.6 | 0 |
| pontepatrullar_biblio.wav | oye pibot bonda patrullar | patrol | 0.4 | 1 |
| ponte_patrullar_office.wav | oye pibot aca patrullar | patrol | 0.4 | 1 |
| sabes_evento_biblio.wav | oye pibot que moes de le ven | | 0.67 | 0 |
| sabes_evento_laboratorio.wav | oye pibot que sabes de evento | talk_about_event | 0.17 | 1 |
| sabes_evento_office.wav | oye pibot que sabes de evento | talk_about_event | 0.17 | 1 |
| ven_aca_office.wav | oye pibot ven par aca | come_towards_me | 0.2 | 1 |
| ven_biblio.wav | oye pibot clelac | talk_about_event | 0.6 | 0 |
| ven_laboratorio.wav | oye pibot ven par aca | come_towards_me | 0.2 | 1 |
| voltea_biblio.wav | oye pibot voltea | look_at_me | 0 | 1 |
| voltea_laboratorio.wav | oye pibot voltea | look_at_me | 0 | 1 |
| voltea_office.wav | oye pibot voltea | look_at_me | 0 | 1 |

TABLE V

LIST OF GENERATED AUDIO FILES WITH THEIR TRANSCRIPTION, INFERRED TEXT, DETECTED COMMAND, WER, AND COMMAND ACCURACY METRICS.