# OpenReview forum: "Work in Progress: Enhancing Human-Robot Interaction through a Speech and Command Recognition System for a Service Robot Using ROS Melodic"
_IEEE.org/ICIST/2024/Conference — IEEE ICIST 2024 Conference Submission_

### Official Review · Reviewer_wZ9Q · 2024-08-21
**This manuscript has a certain degree of innovation and clear simulation figures. It is recommended to accept this paper for publication.**

**Rating:** 7
**Confidence:** 4

**Review:**

This manuscript has a certain degree of innovation and clear simulation figures. Please reply to the following review questions.

Can you elaborate on the specific hardware and software architecture used to integrate the Speech and Command Recognition system into PiBot? How does the system leverage the ROS Melodic framework and the Jetson TX2 embedded computer to process speech inputs and execute commands?

How did you evaluate the system's performance across different testing environments? Were there any particular factors (e.g., background noise levels, distance from the microphone, accent variations) that had a significant impact on the Word Error Rate (WER) and Command Accuracy? Could you provide more details on the experimental setup and analysis of these factors?

---

### Official Review · Reviewer_awWE · 2024-08-21
**This paper presents the development and evaluation of a Speech and Command Recognition system integrated into PiBot. This paper presents an interesting approach. However, the following comments should be considered in the revision.**

**Rating:** 7
**Confidence:** 3

**Review:**

Question 1:
Can the authors elaborate on the specific challenges encountered in implementing speech recognition on the Jetson TX2 embedded computer within real-world environments? How did these hardware constraints influence system performance?
Question 2:
Regarding the achieved 25% Word Error Rate (WER) and 73% Command Accuracy, could the authors provide insights into how performance varied across different testing environments? What were the key factors influencing these variations?
Question 3:
The abstract mentions difficulties in recognizing uncommon or non-Spanish words, indicating a need for further model fine-tuning. Could the authors outline their approach to fine-tuning and improving model performance in diverse acoustic scenarios? What specific strategies will be employed in future work to enhance the system’s reliability in complex, noise-prone settings?

---

### Official Review · Reviewer_a2vi · 2024-08-22
**Manuscript Accept**

**Rating:** 7
**Confidence:** 3

**Review:**

This work is novel. Some issues should be considered in the revision.
How do the authors address the difficulties in recognizing uncommon or non-Spanish words?
How do the authors obtain and utilize the datasets with ground truth transcriptions?
What is the main difficulty in the prospect of Command Accuracy?

---

### Decision · Program_Chairs · 2024-09-08

Accept (Oral)